# Lookahead Bayesian Optimization with Inequality Constraints

**Remi R. Lam**
Massachusetts Institute of Technology
Cambridge, MA
rlam@mit.edu

**Karen E. Willcox**
Massachusetts Institute of Technology
Cambridge, MA
kwillcox@mit.edu

## Abstract

We consider the task of optimizing an objective function subject to inequality constraints when both the objective and the constraints are expensive to evaluate. Bayesian optimization (BO) is a popular way to tackle optimization problems with expensive objective function evaluations, but has mostly been applied to unconstrained problems. Several BO approaches have been proposed to address expensive constraints but are limited to greedy strategies maximizing immediate reward. To address this limitation, we propose a lookahead approach that selects the next evaluation in order to maximize the long-term feasible reduction of the objective function. We present numerical experiments demonstrating the performance improvements of such a lookahead approach compared to several greedy BO algorithms, including constrained expected improvement (EIC) and predictive entropy search with constraint (PESC).

## 1 Introduction

Constrained optimization problems are often challenging to solve, due to complex interactions between the goals of minimizing (or maximizing) the objective function while satisfying the constraints. In particular, non-linear constraints can result in complicated feasible spaces, sometimes partitioned in disconnected regions. Such feasible spaces can be difficult to explore for a local optimizer, potentially preventing the algorithm from converging to a global solution. Global optimizers, on the other hand, are designed to tackle disconnected feasible spaces and optimization of multi-modal objective functions. Such algorithms typically require a large number of evaluations to converge. This can be prohibitive when the evaluation of the objective function or the constraints is expensive, or when there is a *finite budget* of evaluations allocated for the optimization, as it is often the case with expensive models. This evaluation budget typically results from resource scarcity such as the restricted availability of a high-performance computer, finite financial resources to build prototypes, or even time when working on a paper submission deadline.

Bayesian optimization (BO) [19] is a global optimization technique designed to address problems with expensive function evaluations. Its constrained extension, constrained Bayesian optimization (CBO), iteratively builds a statistical model for the objective function and the constraints. Based on this model that leverages all the past evaluations, a *utility function* quantifies the merit of evaluating any design under consideration. At each iteration, a CBO algorithm evaluates the expensive objective function and constraints at the design which maximizes this utility function.

In most existing methods, the utility function only quantifies the reward obtained over the immediate next step, and ignores the gains that could be collected at future steps. This results in greedy CBO algorithms. However, quantifying long-term rewards may be beneficial. For instance, in the presence of constraints, it could be valuable to learn the boundaries of the feasible space. In order to do so, it is likely that an infeasible design would need to be evaluated, bringing no immediate improvement,

but leading to long-term benefits. Such strategy requires planning over several steps. Planning is also required to balance the so-called *exploration-exploitation* trade-off. Intuitively, in order to improve the statistical model, the beginning of the optimization should mainly be dedicated to exploring the design space, while the end of the optimization should focus on exploiting that statistical model to find the best design. To balance this trade-off in a principled way, the optimizer needs to plan ahead and be aware of the remaining evaluation budget.

To address the shortcomings of greedy algorithms, we propose a new *lookahead* formulation for CBO with a finite budget. This approach is aware of the remaining budget and can balance the exploration-exploitation trade-off in a principled way. In this formulation, the best optimization policy sequentially evaluates the design yielding the maximum cumulated reward over multiple steps. This optimal policy is the solution of an intractable dynamic programming (DP) problem. We circumvent this issue by employing an approximate dynamic programming (ADP) algorithm: rollout, building on the unconstrained BO algorithm in [17]. Numerical examples illustrate the benefits of the proposed lookahead algorithm over several greedy ones, especially when the objective function is multi-modal and the feasible space has a complex topology.

The next section gives an overview of CBO and discusses some of the related work (Sec. 2). Then, we formulate the lookahead approach to CBO as a dynamic programming problem and demonstrate how to approximately solve it by adapting the rollout algorithm (Sec. 3). Numerical results are provided in Sec. 4. Finally, we present our conclusions in Sec. 5.

## 2 Constrained Bayesian Optimization

We consider the following optimization problem:

$$
\text{(OPc)} \quad \boldsymbol{x}^* = \underset{\boldsymbol{x} \in \mathcal{X}}{\operatorname{argmin}} f(\boldsymbol{x})
$$
$$
\text{s.t.} \quad g_i(\boldsymbol{x}) \leq 0, \forall i \in \{1, \ldots, I\},
\tag{1}
$$

where $\boldsymbol{x}$ is a $d$-dimensional vector of design variables. The design space $\mathcal{X}$ is a bounded subset of $\mathbb{R}^d$, $f : \mathcal{X} \mapsto \mathbb{R}$ is an objective function, $I$ is the number of inequality constraints and $g_i : \mathcal{X} \mapsto \mathbb{R}$ is the $i^{th}$ constraint function. The functions $f$ and $g_i$ are considered expensive to evaluate. We are interested in finding the minimizer $\boldsymbol{x}^*$ of the objective function $f$ subject to the non-linear constraints $g_i \leq 0$ with a finite budget of $N$ evaluations. We refer to this problem as the *original constrained problem* (OPc).

Constrained Bayesian optimization (CBO) addresses the original constrained problem (OPc) by modeling the objective function $f$ and the constraints $g_i$ as realizations of stochastic processes. Typically, each expensive-to-evaluate function is modeled with an independent Gaussian process (GP). At every iteration $n$, new evaluations of $f$ and $g_i$ become available and augment a training set $\mathcal{S}_n = \{(\boldsymbol{x}_j, f(\boldsymbol{x}_j), g_1(\boldsymbol{x}_j), \cdots, g_I(\boldsymbol{x}_j))\}_{j=1}^n$. Using Bayes rule, the statistical model is updated and the posterior quantities of the GP, conditioned on $\mathcal{S}_n$, reflect the current representation of the unknown expensive functions. In particular, for any design $\boldsymbol{x}$, the posterior mean $\overline{\mu}_n(\boldsymbol{x}; \varphi)$ and the posterior variance $\overline{\sigma}_n^2(\boldsymbol{x}; \varphi)$ of the GP associated with the expensive function $\varphi \in \{f, g_1, \cdots, g_I\}$ can be computed cheaply using a closed-form expression (see [24] for an overview of GP). CBO leverages this statistical model to quantify, in a cheap-to-evaluate utility function $U_n$, the usefulness of any design under consideration. The next design to evaluate is then selected by solving the following auxiliary problem (AP):

$$
\text{(AP)} \quad \boldsymbol{x}_{n+1} = \underset{\boldsymbol{x} \in \mathcal{X}}{\operatorname{argmax}} U_n(\boldsymbol{x}; \mathcal{S}_n).
\tag{2}
$$

The vanilla CBO algorithm is summarized in Algorithm 1.

Many utility functions have been proposed in the literature. To decide which design to evaluate next, [27] proposed the use of constrained expected improvement $EI_c$, which, in the case of independent GPs, can be computed in closed-form as the product of the expected improvement (obtained by considering the GP associated with the objective function) and the probability of feasibility associated with each constraint. This approach was later applied to machine learning applications [6] and extended to the multi-objective case [5]. Note that this method transforms an original constrained optimization problem into an unconstrained auxiliary problem by modifying the utility function. Other attempts to cast the constrained problem into an unconstrained one include [3]. That work uses

---

**Algorithm 1** Constrained Bayesian Optimization

---
    **Input:** Initial training set $\mathcal{S}_1$, budget $N$
    **for** $n = 1$ **to** $N$ **do**
        Construct GPs using $\mathcal{S}_n$
        Update hyper-parameters
        Solve AP for $\boldsymbol{x}_{n+1} = \operatorname{argmax}_{\boldsymbol{x} \in \mathcal{X}} U_n(\boldsymbol{x}; \mathcal{S}_n)$
        Evaluate $f(\boldsymbol{x}_{n+1}), g_1(\boldsymbol{x}_{n+1}), \cdots, g_I(\boldsymbol{x}_{n+1})$
        $\mathcal{S}_{n+1} = \mathcal{S}_n \cup \{(\boldsymbol{x}_{n+1}, f(\boldsymbol{x}_{n+1}), g_1(\boldsymbol{x}_{n+1}), \cdots, g_I(\boldsymbol{x}_{n+1}))\}$
    **end for**

---

a penalty method to transform the original constrained problem into an unconstrained problem, to which they apply a radial basis functions (RBF) method for global optimization (constrained RBF methods exist as well [25]). Other techniques from local constrained optimization have been leveraged in [10] where the utility function is constructed based on an augmented Lagrangian formulation. This technique was recently extended in [22] where a slack-variables formulation allows the handling of equality and mixed constraints. Another approach is proposed by [1]: at each iteration, a finite set of candidate designs is first generated from a Latin hypercube, second, candidate designs with expected constraint violation higher than a user-defined threshold are rejected. Finally, among the remaining candidates, the ones achieving the best expected improvement are evaluated (several designs can be selected simultaneously at each iteration in this formulation). Another method [26] solves a constrained auxiliary optimization problem: the next design is selected to maximize the expected improvement subject to approximated constraints (the posterior mean of the GP associated with a constraint is used in lieu of the constraint itself). Note that the two previous methods solve a constrained auxiliary problem.

Another method to address constrained BO is proposed by [11], who develop an integrated conditional expected improvement criterion. Given a candidate design, this criterion quantifies the expected improvement point-wise (conditioned on the fact that the candidate will be evaluated). This point-wise improvement is then integrated over the entire design space. In the unconstrained case, in the integration phase, equal weight is given to designs throughout the design space. The constrained case is addressed by defining a weight function that depends on the feasible probability of a design: improvement at designs that are likely to be infeasible have low weight. The probability of a design being feasible is calculated using a classification GP. The computation of this criterion is more involved as there is no closed-form formulation available for the integration and techniques such as Monte Carlo or Markov chain Monte Carlo must be employed. In a similar spirit, [21] introduces a utility function which quantifies the benefit of evaluating a design by integrating its effect over the design space. The proposed utility function computes the expected reduction of the feasible domain below the best feasible value evaluated so far. This results in the expected volume of excursion criteria which also requires approximation techniques to be computed.

The former approaches revolve around computing a quantity based on improvement and require having at least one feasible design. Other strategies use information gain as the key element to drive the optimization strategy. [7] proposed a two-step approach for constrained BO when the objective and the constraints can be evaluated independently. The first step chooses the next location by maximizing the constrained EI [27], the second step chooses whether to evaluate the objective or a constraint using an information gain metric (i.e., entropy search [12]). [13, 14] developed a strategy that simultaneously selects the design to be evaluated and the model to query (the objective or a constraint). The criterion used, predictive entropy search with constraints (PESC), is an extension of predictive entropy search (PES) [15]. One of the advantages of information gain-based methods stems from the fact that one does not need to start with a feasible design.

All aforementioned methods use myopic utilities to select the next design to evaluate, leading to suboptimal optimization strategies. In the unconstrained BO setting, multiple-steps lookahead algorithms have been explored [20, 8, 18, 9, 17] and were shown to improve the performance of BO. To our knowledge, such lookahead strategies for constrained optimization have not yet been addressed in the literature and also have the potential to improve the performance of CBO algorithms.

# 3 Lookahead Formulation of CBO

In this section, we formulate CBO with a finite budget as a dynamic programming (DP) problem (Sec. 3.1). This leads to an optimal but computationally challenging optimization policy. To mitigate the cost of computing such a policy, we employ an approximate dynamic programming algorithm, rollout, and demonstrate how it can be adapted to CBO with a finite budget (Sec. 3.2).

## 3.1 Dynamic Programming Formulation

We seek an optimization policy which leads, after consumption of the evaluation budget, to the maximum feasible decrease of the objective function. Because the value of the expensive objective function and constraints are not known before their evaluations, it is impossible to quantify such long-term reward within a cheap-to-evaluate utility function $U_n$. However, CBO endows the objective function and the constraints with a statistical model that can be interrogated to inform the optimizer of the likely values of $f$ and $g_i$ at a given design. This statistical model can be leveraged to simulate optimization scenarios over multiple steps and quantify their probabilities. Using this simulation mechanism, it is possible to quantify, in an average sense, the long-term reward achieved under a given optimization policy. The optimal policy is the solution of the DP problem that we formalize now.

Let $n$ be the current iteration number of the CBO algorithm, and $N$ the total budget of evaluations, or *horizon*. We refer to the future iterations of the optimization generated by simulation as *stages*. For any stage $k \in \{n, \cdots, N\}$, all the information collected is contained in the training set $\mathcal{S}_k$. The function $f$ and the $I$ functions $g_i$ are modeled with independent GPs. Their posterior quantities, conditioned on $\mathcal{S}_k$, fully characterize our knowledge of $f$ and $g_i$. Thus, we define the *state* of our knowledge at stage $k$ to be the training set $\mathcal{S}_k \in \mathcal{Z}_k$.

Based on the training set $\mathcal{S}_k$, the simulation makes a *decision* regarding the next design $\boldsymbol{x}_{k+1} \in \mathcal{X}$ to evaluate using an optimization policy. An optimization policy $\boldsymbol{\pi} = \{\pi_1, \cdots, \pi_N\}$ is a sequence of rules, $\pi_k : \mathcal{Z}_k \mapsto \mathcal{X}$ for $k \in \{1, \cdots, N\}$, mapping a training set $\mathcal{S}_k$ to a design $\boldsymbol{x}_{k+1} = \pi_k(\mathcal{S}_k)$.

In the simulations, the values $f(\boldsymbol{x}_{k+1})$ and $g_i(\boldsymbol{x}_{k+1})$ are unknown and are treated as *uncertainties*. We model those $I + 1$ uncertain quantities with $I + 1$ independent Gaussian random variables $W_{k+1}^f$ and $W_{k+1}^{g_i}$ based on the GPs:

$$W_{k+1}^f \sim \mathcal{N}(\overline{\mu}_k(\boldsymbol{x}_{k+1}; f), \overline{\sigma}_k^2(\boldsymbol{x}_{k+1}; f)), \tag{3}$$

$$W_{k+1}^{g_i} \sim \mathcal{N}(\overline{\mu}_k(\boldsymbol{x}_{k+1}; g_i), \overline{\sigma}_k^2(\boldsymbol{x}_{k+1}; g_i)), \tag{4}$$

where we recall that $\overline{\mu}_k(\boldsymbol{x}_{k+1}; \varphi)$ and $\overline{\sigma}_k^2(\boldsymbol{x}_{k+1}; \varphi)$ are the posterior mean and variance of the GP associated with any expensive function $\varphi \in \{f, g_1, \cdots, g_I\}$, conditioned on $\mathcal{S}_k$, at $\boldsymbol{x}_{k+1}$. Then, the simulation generates an outcome. A simulated outcome $\boldsymbol{w}_{k+1} = (f_{k+1}, g_{k+1}^1, \cdots, g_{k+1}^I) \in \mathcal{W} \subset \mathbb{R}^{I+1}$ is a sample of the $(I+1)$-dimensional random variable $W_{k+1} = [W_{k+1}^f, W_{k+1}^{g_1}, \cdots, W_{k+1}^{g_I}]$. Note that simulating an outcome does not require evaluating the expensive $f$ and $g_i$. In particular, $f_{k+1}$ and $g_{k+1}^i$ are not $f(\boldsymbol{x}_{k+1})$ and $g_i(\boldsymbol{x}_{k+1})$.

Once an outcome $\boldsymbol{w}_{k+1} = (f_{k+1}, g_{k+1}^1, \cdots, g_{k+1}^I)$ is simulated, the system transitions to a new state $\mathcal{S}_{k+1}$, governed by the system dynamic $\mathcal{F}_k : \mathcal{Z}_k \times \mathcal{X} \times \mathcal{W} \mapsto \mathcal{Z}_{k+1}$ given by:

$$\mathcal{S}_{k+1} = \mathcal{F}_k(\mathcal{S}_k, \boldsymbol{x}_{k+1}, \boldsymbol{w}_{k+1}) = \mathcal{S}_k \cup \{(\boldsymbol{x}_{k+1}, f_{k+1}, g_{k+1}^1, \cdots, g_{k+1}^I))\}. \tag{5}$$

Now that the simulation mechanism is defined, one needs a metric to assert the quality of a given optimization policy. At stage $k$, a stage-reward function $r_k : \mathcal{Z}_k \times \mathcal{X} \times \mathcal{W} \mapsto \mathbb{R}$ quantifies the merit of querying a design if the outcome $\boldsymbol{w}_k = (f_{k+1}, g_{k+1}^1, \cdots, g_{k+1}^I)$ occurs. This stage-reward is defined as the reduction of the objective function satisfying the constraints:

$$r_k(\mathcal{S}_k, \boldsymbol{x}_{k+1}, \boldsymbol{w}_{k+1}) = \max\left\{0, f_{best}^{\mathcal{S}_k} - f_{k+1}\right\}, \tag{6}$$

if $g_{k+1}^i \leq 0$ for all $i \in \{1, \cdots, I\}$, and $r_k(\cdot, \cdot, \cdot) = 0$ otherwise, where $f_{best}^{\mathcal{S}_k}$ is the best feasible value at stage $k$. Thus, the expected (long-term) reward starting from training set $\mathcal{S}_n$ under optimization policy $\boldsymbol{\pi}$ is:

$$J_{\boldsymbol{\pi}}(\mathcal{S}_n) = \mathbb{E}\left[\sum_{k=n}^{N} r_k(\mathcal{S}_k, \pi_k(\mathcal{S}_k), \boldsymbol{w}_{k+1})\right], \tag{7}$$

where the expectation is taken with respect to the (correlated) simulated values $(\boldsymbol{w}_{n+1}, \cdots, \boldsymbol{w}_{N+1})$, and the state evolution is governed by Eq. 5. An optimal policy, $\boldsymbol{\pi}^*$, is a policy maximizing this long-term expected reward in the space of admissible policies $\Pi$:

$$J_{\boldsymbol{\pi}^*}(\mathcal{S}_n) = \max_{\boldsymbol{\pi} \in \Pi} J_{\boldsymbol{\pi}}(\mathcal{S}_n). \tag{8}$$

The optimal reward $J_{\boldsymbol{\pi}^*}(\mathcal{S}_n)$ is given by Bellman's principle of optimality and can be computed using the DP recursive algorithm, working backward from $k = N - 1$ to $k = n$,

$$J_N(\mathcal{S}_N) = \max_{\boldsymbol{x}_{N+1} \in \mathcal{X}} \mathbb{E}[r_N(\mathcal{S}_N, \boldsymbol{x}_{N+1}, \boldsymbol{w}_{N+1})] = \max_{\boldsymbol{x}_{N+1} \in \mathcal{X}} EI_c(\boldsymbol{x}_{N+1}; \mathcal{S}_N)$$

$$J_k(\mathcal{S}_k) = \max_{\boldsymbol{x}_{k+1} \in \mathcal{X}} \mathbb{E}[r_k(\mathcal{S}_k, \boldsymbol{x}_{k+1}, \boldsymbol{w}_{k+1}) + J_{k+1}(\mathcal{F}_k(\mathcal{S}_k, \boldsymbol{x}_{k+1}, \boldsymbol{w}_{k+1}))], \tag{9}$$

where each expectation is taken with respect to one simulated outcome vector $\boldsymbol{w}_{k+1}$, and we have used the fact that $\mathbb{E}[r_k(\mathcal{S}_k, \boldsymbol{x}_{k+1}, \boldsymbol{w}_{k+1})] = EI_c(\boldsymbol{x}_{k+1}; \mathcal{S}_k)$ is the constrained expected improvement known in closed-form [27]. The optimal reward is given by $J_{\boldsymbol{\pi}^*}(\mathcal{S}_n) = J_n(\mathcal{S}_n)$. Thus, at iteration $n$ of the CBO algorithm, the optimal policy select the next design $\boldsymbol{x}_{n+1}$ that maximizes $J_n(\mathcal{S}_n)$ given by Eqs. 9. In other words, the best decision to make at iteration $n$ maximizes, on average, the sum of the immediate reward $r_n$ and the future long-term reward $J_{n+1}(\mathcal{S}_{n+1})$ obtained by making optimal subsequent decisions. This is illustrated in Fig. 1, left panel.

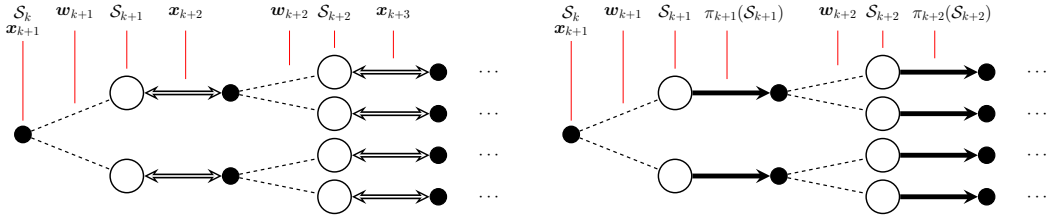

Figure 1: Left: Tree illustrating the intractable DP formulation. Each black circle represents a training set and a design, each white circle is a training set. Dashed lines represent simulated outcomes resulting in expectations. The double arrows represent designs selected with the (unknown) optimal policy, leading to nested maximizations. Double arrows depict the bidirectional way information propagates when the optimal policy is built: each optimal decision depends on the previous steps and relies on the optimality of the future decisions. Right: Single arrows represent designs selected using a heuristic. This illustrates the unidirectional propagation of information when a known heuristic drives the simulations: each decision depends on the previous steps but is independent of the future ones. The absence of nested maximization leads to a tractable formulation.

## 3.2 Rollout for Constrained Bayesian Optimization

The best optimization policy evaluates, at each iteration $n$ of the CBO algorithm, the design $\boldsymbol{x}_{n+1}$ maximizing the optimal reward $J_{\boldsymbol{\pi}^*}(\mathcal{S}_n)$ (Eq. 8). This requires solving a problem with several nested maximizations and expectations (Eqs. 9), which is computationally intractable. To mitigate the cost of solving the DP algorithm, we employ an approximate dynamic programming (ADP) technique: rollout (see [2, 23] for an overview). Rollout selects the next design by maximizing a (suboptimal) long-term reward $J_{\boldsymbol{\pi}}$. The reward is computed by simulating optimization scenarios over several future steps. However, the simulated steps are not controlled by the optimal policy $\boldsymbol{\pi}^*$. Instead, rollout uses a suboptimal policy $\boldsymbol{\pi}$, i.e. a heuristic, to drive the simulation. This circumvents the need for nested maximizations (as illustrated in Fig. 1, right panel) and simplifies the computation of $J_{\boldsymbol{\pi}}$ compared to $J_{\boldsymbol{\pi}^*}$. We now formalize the rollout algorithm, propose a heuristic $\boldsymbol{\pi}$ adapted to the context of CBO with a finite budget, and detail further numerical approximations.

Let us consider the iteration $n$ of the CBO algorithm. The long-term reward $J_{\boldsymbol{\pi}}(\mathcal{S}_n)$ induced by a (known) heuristic $\boldsymbol{\pi} = \{\pi_1, \cdots, \pi_N\}$, starting from state $\mathcal{S}_n$, is defined by Eq. 7. This can be rewritten as $J_{\boldsymbol{\pi}}(\mathcal{S}_n) = H_n$, where $H_n$ is recursively defined, from $k = N$ back to $k = n$, by:

$$H_{N+1}(\mathcal{S}_{N+1}) = 0$$

$$H_k(\mathcal{S}_k) = \mathbb{E}[r_k(\mathcal{S}_k, \pi_k(\mathcal{S}_k), \boldsymbol{w}_{k+1}) + \gamma H_{k+1}(\mathcal{F}_k(\mathcal{S}_k, \pi_k(\mathcal{S}_k), \boldsymbol{w}_{k+1}))], \tag{10}$$

where each expectation is taken with respect to one simulated outcome vector $\boldsymbol{w}_{k+1}$, and $\gamma \in [0, 1]$ is a discount factor encouraging the early collection of reward. A discount factor $\gamma = 0$ leads to a greedy policy, focusing on immediate reward. In that case, the reward $J_{\boldsymbol{\pi}}$ simplifies to the constrained expected improvement $EI_c$. A discount factor $\gamma = 1$, on the other hand, is indifferent to when the reward is collected.

The fundamental simplification introduced by the rollout algorithm lies in the absence of nested maximizations in Eqs. 10. This is illustrated in Fig. 1, right panel. By applying a known heuristic, information only propagates forward: every simulated step depends on the previous steps, but is independent from the future simulated steps. This is in contrast to the DP algorithm, illustrated in Fig. 1. Because the optimal policy is not known, it needs to be built by solving a sequence of nested problems. Thus, information propagates both forward and backward.

While $H_n$ is simpler to compute than $J_n$, it still requires computing nested expectations for which there is no closed-form expression. To further alleviate the cost of computing the long-term reward, we introduce two numerical simplifications. First, we use a rolling horizon $h \in \mathbb{N}$ to decrease the number of future steps simulated. A *rolling horizon* $h$ replaces the horizon $N$ by $\tilde{N} = \min\{N, n+h\}$. Second, the expectations with respect to the $(I + 1)$-dimensional Gaussian random variables are numerically approximated using Gauss-Hermite quadrature. We obtain the following formulation:

$$
\begin{aligned}
\tilde{H}_{\tilde{N}+1}(\mathcal{S}_{\tilde{N}+1}) &= 0 \\
\tilde{H}_k(\mathcal{S}_k) &= EI_c(\pi_k(\mathcal{S}_k); \mathcal{S}_k) + \gamma \sum_{q=1}^{N_q} \alpha^{(q)}[\tilde{H}_{k+1}(\mathcal{F}_k(\mathcal{S}_k, \pi_k(\mathcal{S}_k), \boldsymbol{w}_{k+1}^{(q)}))],
\end{aligned}
\tag{11}
$$

where $N_q$ is the number of quadrature weights $\alpha^{(q)} \in \mathbb{R}$ and points $\boldsymbol{w}_{k+1}^{(q)} \in \mathbb{R}^{I+1}$.

For all iteration $n \in \{1, \cdots, N\}$ and for all $\boldsymbol{x}_{n+1} \in \mathcal{X}$, we define the utility function of our rollout algorithm for CBO with finite budget to be:

$$
U_n(\boldsymbol{x}_{n+1}; \mathcal{S}_n) = EI_c(\boldsymbol{x}_{n+1}; \mathcal{S}_n) + \gamma \sum_{q=1}^{N_q} \alpha^{(q)}[\tilde{H}_{n+1}(\mathcal{F}_n(\mathcal{S}_n, \boldsymbol{x}_{n+1}, \boldsymbol{w}_{n+1}^{(q)}))].
\tag{12}
$$

The heuristic $\boldsymbol{\pi}$ is problem-dependent. A desirable heuristic combines two properties: (1) it is cheap to compute, (2) it is a good approximation of the optimal policy $\boldsymbol{\pi}^*$. In the case of CBO with a finite budget, the heuristic $\boldsymbol{\pi}$ ought to mimic the exploration-exploitation trade-off balanced by the optimal policy $\boldsymbol{\pi}^*$. To do so, we propose using a combination of greedy CBO algorithms: maximization of the constrained expected improvement (which has an exploratory behavior) and a constrained optimization based on the posterior means of the GPs (which has an exploitative behavior). For a given iteration $n$, we define the heuristic $\boldsymbol{\pi} = \{\pi_{n+1}, \cdots, \pi_{\tilde{N}}\}$ such that for stages $k \in \{n+1, \cdots, \tilde{N}-1\}$, the policy component $\pi_k : \mathcal{Z}_k \mapsto \mathcal{X}$, maps a state $\mathcal{S}_k$ to the design $\boldsymbol{x}_{k+1}$ satisfying:

$$
\boldsymbol{x}_{k+1} = \operatorname*{argmax}_{\boldsymbol{x} \in \mathcal{X}} EI_c(\boldsymbol{x}; \mathcal{S}_k).
\tag{13}
$$

The last policy component, $\pi_{\tilde{N}} : \mathcal{Z}_{\tilde{N}} \mapsto \mathcal{X}$, maps a state $\mathcal{S}_{\tilde{N}}$ to $\boldsymbol{x}_{\tilde{N}+1}$ such that:

$$
\boldsymbol{x}_{\tilde{N}+1} = \operatorname*{argmin}_{\boldsymbol{x} \in \mathcal{X}} \overline{\mu}_{\tilde{N}}(\boldsymbol{x}; f) \quad \text{s.t.} \quad PF(\boldsymbol{x}; \mathcal{S}_{\tilde{N}}) \geq 0.99,
\tag{14}
$$

where $PF$ is the probability of feasibility known in closed-form. Every evaluation of the utility function $U_n$ requires $\mathcal{O}\left(N_q^h\right)$ applications of a heuristic component $\pi_k$. The heuristic that we propose optimizes a quantity that requires $\mathcal{O}\left(|\mathcal{S}_k|^2\right)$ of work.

To summarize, the proposed approach sequentially selects the next design to evaluate by maximizing the long-term reward induced by a heuristic. This rollout algorithm is a one-step lookahead formulation (one maximization) and is easier to solve than the $N$-steps lookahead approach ($N$ nested maximizations) presented in Sec. 3.1. Rollout is a closed-loop approach where the information collected at a given stage of the simulation is used to simulate the next stages. The heuristic used in the rollout is problem-dependent, and we proposed using a combination of greedy CBO algorithms to construct such a heuristic. The computation of the utility function is detailed in Algorithm 2.

---
**Algorithm 2** Rollout Utility Function
---
**Function:** `utility`$(\boldsymbol{x}, h, \mathcal{S})$
Construct GPs using $\mathcal{S}$
**if** $h = 0$ **then**
   $U \leftarrow EI_c(\boldsymbol{x}; \mathcal{S})$
**else**
   $U \leftarrow EI_c(\boldsymbol{x}; \mathcal{S})$
   Generate $N_q$ Gauss-Hermite quadrature weights $\alpha^{(q)}$ and points $\boldsymbol{w}^{(q)}$ associated with $\boldsymbol{x}$
   **for** $q = 1$ **to** $N_q$ **do**
      $\mathcal{S}' \leftarrow \mathcal{S} \cup \{(\boldsymbol{x}, \boldsymbol{w}^{(q)})\}$
      **if** $h > 1$ **then**
         $\boldsymbol{x}' \leftarrow \pi(\mathcal{S}')$ using Eq. 13
      **else**
         $\boldsymbol{x}' \leftarrow \pi(\mathcal{S}')$ using Eq. 14
      **end if**
      $U \leftarrow U + \gamma \alpha^{(q)}$`utility`$(\boldsymbol{x}', h - 1, \mathcal{S}')$
   **end for**
**end if**
**Output:** $U$
---

## 4 Results

In this section, we numerically investigate the proposed algorithm and demonstrate its performance on classic test functions and a reacting flow problem.

To compare the performance of the different CBO algorithms tested, we use the utility gap metric [14]. At iteration $n$, the utility gap $e_n$ measures the error between the optimum feasible value $f^*$ and the value of the objective function at a recommended design $\boldsymbol{x}_n^*$:

$$e_n = \begin{cases} |f(\boldsymbol{x}_n^*) - f^*| & \text{if } \boldsymbol{x}_n^* \text{ is feasible,} \\ |\Psi - f^*| & \text{else,} \end{cases} \tag{15}$$

where $\Psi$ is a user-defined penalty punishing infeasible recommendations. The recommended design, $\boldsymbol{x}_n^*$, differs from the design selected for evaluation $\boldsymbol{x}_n$. It is the design that the algorithm would recommend to evaluate if the optimization were to be stopped at iteration $n$, without early notice. We use the same system of recommendation as [14]:

$$\boldsymbol{x}_n^* = \operatorname*{argmin}_{\boldsymbol{x} \in \mathcal{X}} \overline{\mu}_n(\boldsymbol{x}; f) \quad \text{s.t.} \quad PF(\boldsymbol{x}; \mathcal{S}_n) \geq 0.975. \tag{16}$$

Note that the utility gap $e_n$ is not guaranteed to decrease because recommendations $\boldsymbol{x}_n^*$ are not necessarily better with iterations. In particular, $e_n$ is not the best error achieved in the training set $\mathcal{S}_n$.

In the following numerical experiments, for the rollout algorithm, we use independent zero-mean GPs with automatic relevance determination (ARD) square-exponential kernel to model each expensive-to-evaluate function. In Algorithm. 1, when the GPs are constructed, the vector of hyper-parameters $\boldsymbol{\theta}_i$ associated with the $i^{th}$ GP kernel is estimated by maximization of the marginal likelihood. However, to reduce the cost of computing $U_n$, the hyper-parameters are kept constant in the simulated steps (i.e., in Algorithm. 2). To compute the expectations of Eqs. 11-12, we employ $N_q = 3^{I+1}$ Gauss-Hermite quadrature weights and points and we set the discount factor to $\gamma = 0.9$. Finally, at iteration $n$, the best value $f_{best}^{\mathcal{S}_n}$ is set to the minimum posterior mean $\overline{\mu}_n(\boldsymbol{x}; f)$ over the designs $\boldsymbol{x}$ in the training set $\mathcal{S}_n$, such that the posterior mean of each constraint is feasible. If no such point can be found, then $f_{best}^{\mathcal{S}_n}$ is set to the maximum of $\{\overline{\mu}_n(\boldsymbol{x}; f) + 3\sigma_m\}$ over the designs $\boldsymbol{x}$ in $\mathcal{S}_n$, where $\sigma_m^2$ is the maximum variance of the GP associated with $f$. The EIC algorithm is computed as a special case of the rollout with rolling horizon $h = 0$, and we use the Spearmint package[1] to run the PESC algorithm. We additionally run a CBO algorithm that selects the next design to evaluate based on the posterior means of the GPs[2]:

$$\boldsymbol{x}_{n+1} = \operatorname*{argmin}_{\boldsymbol{x} \in \mathcal{X}} \overline{\mu}_n(\boldsymbol{x}; f) \quad \text{s.t.} \quad \overline{\mu}_n(\boldsymbol{x}; g_i) \leq 0, \forall i \in \{1, \dots, I\}. \tag{17}$$

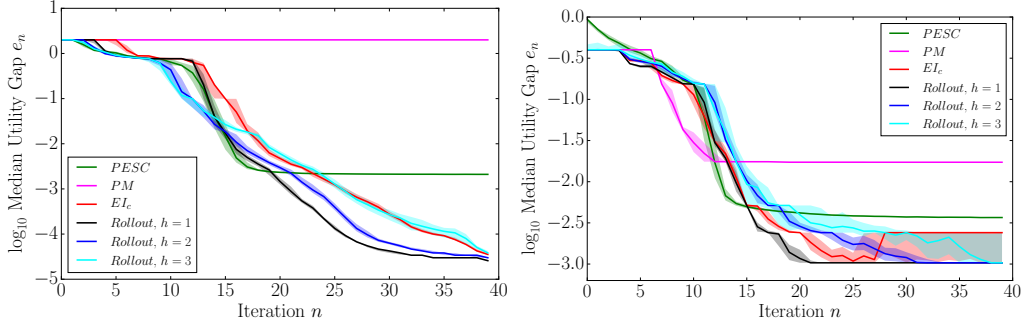

Figure 2: Left: Multi-modal objective and single constraint (P1). Right: Linear objective and multiple non-linear constraints (P2). Shaded region indicates 95% confidence interval of the median statistic.

We refer to this algorithm as PM. We also compare the CBO algorithms to three local algorithms (SLSQP, MMA and COBYLA) and to one global evolutionary algorithm (ISRES).

We now consider four problems with different design space dimensions $d$, several numbers of constraints $I$, and various topologies of the feasible space. The three first problems, P1-3, are analytic functions while the last one, P4, uses a reacting flow model that requires solving a set of partial differential equations (PDEs) [4]. For P1 and P2, we use $N = 40$ evaluations (as in [6, 10]). For P3 and P4, we use a small number of iterations $N = 60$, which corresponds to situations where the functions are very expensive to evaluate (e.g. solving large systems of PDEs can take over a day on a supercomputer). The full description of the problems is available in the appendix. In Figs. 2-3, we show the median of the utility gap, the shadings represent the 95% confidence interval of the median computed by bootstrap. Other statistics of the utility gap are shown in the appendix.

For P1, the median utility gap for EIC, PESC, PM and the rollout algorithm with $h \in \{1, 2, 3\}$ is shown in Fig. 2 (left panel). The PM algorithm does not improve its recommendations. This is not surprising because PM focuses on exploitation (PM does not depends on posterior variance) which can result in the algorithm failing to make further progress. Such behavior has already been reported in [16] (Sec. 3). The three other CBO algorithms perform similarly in the first 10 iterations. PESC is the first to converge to a utility gap $\approx 10^{-2.7}$. The rollout performs better or similarly than EIC. In the 15 first iterations, longer rolling horizons lead to slightly lower utility gaps. This is likely to be due to the more exploratory behavior associated with lookahead, which helps differentiating the global solution from the local ones. For the remaining iterations, the shorter rolling horizons reduce the utility gap faster than longer rolling horizons before reaching a plateau. EIC and rollout outperform PESC after 25 iterations. We note that EIC and rollout have essentially converged.

For P2, the median performance of EIC, PESC, PM and rollout with rolling horizon $h \in \{1, 2, 3\}$ is shown in Fig. 2 (right panel). The PM algorithm reduces the utility gap in the first 10 iterations, but reaches a plateau at $10^{-1.7}$. The three other CBO algorithms perform similarly up to iteration 15, where PESC reaches a plateau [3]. This similarity may be explained by the fact that the local solutions are easily differentiable from the global one, leading to no advantage for exploratory behavior. In this example, the rollout algorithms reached the same plateau at $10^{-3}$, with longer horizons $h$ taking more iterations to converge. EIC performs better than rollout $h = 2$ before its performance slightly decreases, reaching a plateau at a larger utility gap $10^{-2.6}$ (note that the utility gap is not computed with the best value observed so far and thus is not guaranteed to decrease). This increase of the median utility gap can be explained by the fact that a few runs change their recommendation from one local minimum to another one, resulting in the change in median utility function. This is also reflected in the 95% confidence interval of the median, which further indicates that the statistic is sensitive to a few runs.

For P3, the median utility gap for the four CBO algorithms is shown in Fig. 3 (left panel). PM is rapidly outperformed by the other algorithms. The PESC algorithm is outperformed by EIC and rollout after 25 iterations. Again, we note that rollout with $h = 1$ obtains a lower utility gap that EIC at every iteration. The rollout with $h \in \{2, 3\}$ exhibits a different behavior: it starts decreasing the utility gap later in the optimization but achieves a better performance when the evaluation budget

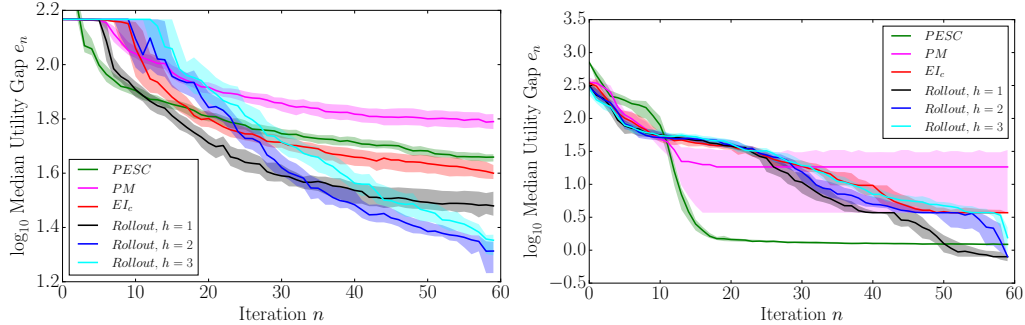

Figure 3: Left: Multi-modal 4-$d$ objective and constraint (P3). Right: Reacting flow problem (P4). The awareness of the remaining budget explains the sharp decrease in the last iterations for the rollout.

is consumed. Note that none of the algorithms has converged to the global solution, and the strong multi-modality of the objective and constraint function seems to favor exploratory behaviors.

For the reacting flow problem P4, the median performances are shown in Fig. 3 (right panel). PM rapidly reaches a plateau at $e_n \approx 10^{1.3}$. PESC reduces rapidly the utility gap, outperforming the other algorithms after 15 iterations. EIC and rollout perform similarly and slowly decrease the utility gap up to iteration 40, where EIC reaches a plateau and rollout continues to improve performance, slightly outperforming PESC at the end of the optimization.

The results are summarized in Table. 1, and show that the rollout algorithm with different rolling horizons $h$ (R-$h$) performs similarly or favorably compared to the other algorithms.

Table 1: Log median utility gap $\log_{10}(e_N)$. Statistics computed over $m$ independent runs.

| Prob | $d$ | $N$ | $I$ | $m$ | SLSQP | MMA | COBYLA | ISRES | PESC | PM | EIC | R-1 | R-2 | R-3 |
|------|-----|-----|-----|-----|-------|-----|--------|-------|------|-----|-----|-----|-----|-----|
| P1 | 2 | 40 | 1 | 500 | *0.59* | *0.59* | -0.05 | -0.19 | -2.68 | 0.30 | -4.45 | **-4.59** | -4.52 | -4.42 |
| P2 | 2 | 40 | 2 | 500 | *-0.40* | *-0.40* | -0.82 | -0.70 | -2.43 | -1.76 | -2.62 | **-2.99** | **-2.99** | -2.99[4] |
| P3 | 4 | 60 | 1 | 500 | 2.15 | *3.06* | *3.06* | 1.68 | 1.66 | 1.79 | 1.60 | 1.48 | **1.31** | 1.35 |
| P4 | 4 | 60 | 1 | 50 | 0.80 | 0.80 | 0.80 | 0.13 | 0.09 | *1.26* | 0.57 | **-0.10** | **-0.10** | 0.19 |

Based on the four previous examples, we notice that increasing the rolling horizon $h$ does not necessarily improve the performance of the rollout algorithm. One possible reason stems from the fact that lookahead algorithms rely more on the statistical model that greedy algorithms. Because this model is learned as the optimization unfolds, it is an imperfect model (in particular the hyper-parameters of the GPs are updated after each iteration, but not after each stage of a simulated scenario). By simulating too many steps with the GPs, one may be over-confidently using the model. In some sense, the rolling horizon $h$, as well as the discount factor $\gamma$, can be interpreted as a form of regularization. The effect of a larger rolling horizon is problem-dependent, and experiment P3 suggests that multimodal problems in higher dimension may benefits from longer rolling horizons.

## 5 Conclusions

We proposed a new formulation for constrained Bayesian optimization with a finite budget of evaluations. The best optimization policy is defined as the one maximizing, in average, the cumulative feasible decrease of the objective function over multiple steps. This optimal policy is the solution of a dynamic programming problem that is intractable due to the presence of nested maximizations. To circumvent this difficulty, we employed the rollout algorithm. Rollout uses a heuristic to simulate optimization scenarios over several step, thereby computing an approximation of the long-term reward. This heuristic is problem-dependent and, in this paper, we proposed to use a combination of cheap-to-evaluate greedy CBO algorithms to construct such heuristic. The proposed algorithm was numerically investigated and performed similarly or favorably compared to constrained expected improvement (EIC) and predictive entropy search with constraint (PESC).

This work was supported in part by the AFOSR MURI on multi-information sources of multi-physics systems under Award Number FA9550-15-1-0038, program manager Dr. Jean-Luc Cambier.

## Footnotes

[1]`https://github.com/HIPS/Spearmint/tree/PESC`

[2]As suggested by a reviewer.

[3]Results obtained for PESC mean utility gap are consistent with [13].

[4]For cost reasons, the median for $h = 3$ was computed with $m = 100$ independent runs instead of 500.

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
