[Supplementary Material · Lam_Willcox_NIPS_2017_Appendix.pdf]

# Lookahead Bayesian Optimization
# with Inequality Constraints
# Appendix

**Remi R. Lam**
Massachusetts Institute of Technology
Cambridge, MA
rlam@mit.edu

**Karen E. Willcox**
Massachusetts Institute of Technology
Cambridge, MA
kwillcox@mit.edu

## P1: Multi-modal Objective and Single Constraint

We first consider the optimization problem proposed in [2] with $d = 2$ design variables on $\mathcal{X} = [0,6]^2$ and $I = 1$ constraint function. The objective function $f$ and the constraint $g$ are defined by:

$$f(\boldsymbol{x}) = \cos(2x_1)\cos(x_2) + \sin(x_1), \tag{1}$$

$$g(\boldsymbol{x}) = \cos(x_1)\cos(x_2) - \sin(x_1)\sin(x_2) + 0.5, \tag{2}$$

where $\boldsymbol{x} = [x_1, \cdots, x_d]^\top$ is a vector of design variables. This problem has five local solutions including a unique global one. The feasible space is composed of two disjoint regions.

We run every algorithm with a budget of $N = 40$ evaluations. Each algorithm is initially given one training point. We repeat 500 times the experiment with independent initial conditions uniformly sampled from the design space (experiments may start without a feasible training point). The penalty value is set to $\Psi = 2$.

Figure 1: Mean (top left), 75% quantile (top right), median (bottom left) and 25% quantiles of the utility gap for P1.

## P2: Linear Objective and Multiple Constraints

We now consider the problem introduced in [3], where the goal is to minimize a linear objective $f$ on $\mathcal{X} = [0,1]^2$ subject to two constraints:

$$f(\boldsymbol{x}) = x_1 + x_2, \tag{3}$$

$$g_1(\boldsymbol{x}) = 0.5\sin(2\pi(2x_2 - x_1^2)) - x_1 - 2x_2 + 1.5, \tag{4}$$

$$g_2(\boldsymbol{x}) = x_1^2 + x_2^2 - 1.5. \tag{5}$$

This problem is characterized by three local solutions including a unique global one. The feasible space is connected. We allocate a budget of $N = 40$ evaluations and repeat the experiment with 500 independent initial conditions for all the algorithms. The penalty value is set to $\Psi = 1$.

Figure 2: Mean (top left), 75% quantile (top right), median (bottom left) and 25% quantiles of the utility gap for P2.

## P3: Multi-modal 4-dimensional Objective and Constraint

We now consider a 4-dimensional problem defined over $\mathcal{X} = [-5, 5]^4$ with one inequality constraint:

$$f(\boldsymbol{x}) = \frac{1}{2} \sum_{i=1}^{d} (x_i^4 - 16x_i^2 + 5x_i), \tag{6}$$

$$g(\boldsymbol{x}) = -0.5 + \sin(x_1 + 2x_2) - \cos(x_3)\cos(2x_4). \tag{7}$$

Both the Styblinski-Tang function[1] $f$ and the constraint $g$ are multi-modal functions. Each algorithm is given a budget of $N = 60$ evaluations and the experiment is repeated 500 times. The penalty value is set to the maximum value of $f$ over $\mathcal{X}$: $\Psi = 1000$.

Figure 3: Mean (top left), 75% quantile (top right), median (bottom left) and 25% quantiles of the utility gap for P3.

**P4: Reacting flow problem**

Finally, we consider the problem of maximizing the heat released by a reacting flow while preventing the maximum temperature from exceeding a threshold $T_{max} = 1800\ K$. We use the reacting flow model described in [1], with the following 4 design variables: equivalence ration $\phi \in [0, 2]$, inlet velocity $u \in [40, 80]$ cm/sec, inlet temperature $T_i \in [850, 1000]\ K$ and wall temperature $T_w \in [200, 400]\ K$. Computing the heat released and the maximum temperature requires numerically solving a system of partial differential equations (PDE). Given the cost of evaluating the objective function and the constraint, we only run the algorithms for 50 independent initial conditions. The penalty value is set to $\Psi = 500\ K$ and the budget is $N = 60$ evaluations.

Figure 4: Mean (top left), 75% quantile (top right), median (bottom left) and 25% quantiles of the utility gap for P4.

## Footnotes

[1] Function from https://www.sfu.ca/~ssurjano/stybtang.html.