[Reviews · NeurIPS 2017]

Reviewer 1



The authors address the problem of Bayesian optimization (BO) with inequality constraints. All BO algorithms for constrained optimization using a myopic strategy in which they just optimize the expected utility of the next immediate evaluation. The authors proposed to consider instead multiple look ahead steps. In this case, computing the optimal next evaluation location requires to solve an intractable dynamic programming problem. To simplify the complexity of the approach, the authors consider a roll out strategy in which a specific policy is applied at each step to select the next evaluation location as a function of just the current state. The proposed policy optimizes the expected improvement with constraints obtained by the next immediate evaluation. Integration over the possible outcomes of each function evaluation is performed numerically using Gauss-Hermite weights. The experiments show that the proposed approach can produce small gains when considering a 1-step look ahead setting.

Reviewer 2



This paper seems a continuation of last year: Bayesian optimization with a finite budget... where the authors have added new elements to deal with inequality constraints. The method uses a approximation of a lookahead strategy by dynamic programming. For the constrained case, the authors propose an heuristic that combines the EIc criterion for all the steps except for the last one were the mean function is used. The authors claim that the mean function has an exploitative behaviour, although it has been previously shown that it might be misleading [A]. A considerably amount of the text, including Figure 1, can be mostly found in [16]. Although it is nice to have an self-contained paper as much as possible, that space could be used to explain better the selection of the acquisition heuristic and present alternatives. For example, the experiments should show the result a single-step posterior mean acquisition function, which will correspond to h=0 as a baseline for each of the functions/experiments. Concerning the experiments, there are some details that should be improved or addressed in the paper: - Why P2 does not include h=3? - How is it possible that EIc for P2 goes upwards around n=25? - The plots only shows median values without error bars. I can understand that for that, for such number of repetitions, the error bars might be small, but that should be adressed. Furthermore, a common problem of lookahead methods is that, while most of the time outperform greedy methods, they can also fail catastrophically for selecting the wrong path. This would results in some lookahead trials resulting in poor results. Using the average instead of the median would show also the robustness of the method. - The number of iterations is fairly small for the standard in Bayesian optimization. In fact, it can bee seen that most methods have not reach convergence at that point. This is specially problematic in P1 and P4 where the plots seems to cross exactly at the end of experiments, - It is unclear why the performance decreases after h=2. If possible, the authors should provide an intuition behind that results. Furthermore, the fact that for most of the experiments h=1 is the optimal strategy seems to indicate that the whole dynamic programing and heuristics might be excessive for CBO, or that more complex and higher-dimensional problems are need to illustrate the benefits of the strategy. [A] Jones, Donald R. "A taxonomy of global optimization methods based on response surfaces." Journal of global optimization 21.4 (2001): 345-383.

Reviewer 3



This work proposes a new method for Bayesian with restrictions in which the policy for collecting new evaluations is none myopic, in the sense that it takes into account the impact of the next evaluation in the future behavior of the policy. Both problems are important in Bayesian optimization and haven't been studied considered together, to the best of my knowledge. The paper is very well written and structured. The problem is interesting and the adaptation of rollout for this scenario is well executed. My only comment is about the experimental section and how the proposed policy has been used. A 'pure' non-myopic policy should ideally consider at each step of the optimization as many look-aheads as the number of remaining evaluations that are available. In their experiments, the authors only consider a maximum of 3 look-ahead steps which makes difficult to evaluate how the policy will perform in the pure non-myopic case. If the reason for this is the computational complexity of the policy, an analysis of the limitations of this method should be described in the paper. If this is not the case, and the policy can be computed in cases with many steps ahead, the authors should include these results in the paper. Apart form this point that I think would improve the quality of this work, I think that this is a good paper.